

# Knowing your neighbourhood—the effects of *Epichloë* endophytes on foliar fungal assemblages in perennial ryegrass in dependence of season and land-use intensity

Julia König[1], Marco Alexandre Guerreiro[2], Derek Peršoh[2], Dominik Begerow[2] and Jochen Krauss[1]

[1] Department of Animal Ecology and Tropical Biology, Biocentre, University of Würzburg, Würzburg, Germany

[2] Department of Geobotany, Ruhr-Universität Bochum, Bochum, Germany

## ABSTRACT

*Epichloë* endophytes associated with cool-season grass species can protect their hosts from herbivory and can suppress mycorrhizal colonization of the hosts' roots. However, little is known about whether or not *Epichloë* endophyte infection can also change the foliar fungal assemblages of the host. We tested 52 grassland study sites along a land-use intensity gradient in three study regions over two seasons (spring vs. summer) to determine whether *Epichloë* infection of the host grass *Lolium perenne* changes the fungal community structure in leaves. Foliar fungal communities were assessed by Next Generation Sequencing of the ITS rRNA gene region. Fungal community structure was strongly affected by study region and season in our study, while land-use intensity and infection with *Epichloë* endophytes had no significant effects. We conclude that effects on non-systemic endophytes resulting from land use practices and *Epichloë* infection reported in other studies were masked by local and seasonal variability in this study's grassland sites.

## INTRODUCTION

Fungi include ubiquitous and highly diverse microbial symbionts associated with a large number of plant species in all terrestrial ecosystems (*Dix & Webster, 1995*). Such associations can have profound effects on ecosystems (*Van der Heijden et al., 1998*; *Clay & Holah, 1999*; *Omacini et al., 2001*; *Bundrett, 2006*). Besides mycorrhizae, some of the best-known plant-fungus interactions are the symbioses of endophytes of the genus *Epichloë* (Ascomycota, Clavicipitaceae) with cool-season grass species in the family Poaceae (*Schardl, Leuchtmann & Spiering, 2004*; *Tanaka et al., 2012*). *Epichloë* endophytes systemically colonize above ground tissues of the host grass. Asexual *Epichloë* species are vertically transmitted through the seeds and provide several benefits to their hosts,

Corresponding author
Julia König,
julia.koenig@uni-wuerzburg.de

like herbivore resistance and enhanced fitness (*Schardl, Leuchtmann & Spiering, 2004*). Sexual species of *Epichloë* endophytes produce spores which are transmitted horizontally by symbiotic flies of the genus *Botanophila* and supress the hosts' seed development (*Schardl, Leuchtmann & Spiering, 2004*). Depending on abiotic and biotic conditions, the asexual *Epichloë* endophyte-grass association can shift from a mutualistic symbiosis to an antagonistic symbiosis, e.g., when herbivore pressure is low and nitrogen availability for the host is limited (*Saikkonen et al., 1998*; *Müller & Krauss, 2005*).

All grass species are associated with a large number of different fungal endophytes and often harbour more than 100 species which colonize roots, stems and leaves of the plants (*Sánchez Márquez, Bills & Zabalgogeazcoa, 2007*). Therefore the systemic *Epichloë* endophytes represent only a small fraction of a diverse fungal community in grass species (*Neubert et al., 2006*). In contrast to *Epichloë* endophytes, most other endophytic fungi have a limited capacity to systemically colonize the plant organs or seeds (*Rodriguez et al., 2009*). In addition, the diverse endophytic fungi have unequal colonization frequencies with a few dominant genera, such as *Alternaria, Acremonium, Cladosporium* and *Penicillium,* which occur in multiple grass species, as well as in non-grass hosts (*Sánchez Márquez et al., 2012*).

Diverse foliar fungal endophytes of plants are influenced by several abiotic and biotic factors which may compromise the host species' ability to colonize, persist and disperse (*Rodriguez et al., 2009*). The fungal assemblages of grass leaves and frequencies of endophyte species change with spatial distance and season (*Sánchez Márquez et al., 2012*). As these fungal species vary in their dispersal ability, dissimilarities between fungal assemblages increase with distance (*David, Seabloom & May, 2016*). Depending on the prevailing microclimate, variability can be high at small spatial scales, e.g., between individual leaves of a single tree, as well as between individual plants or different plant species (*Stone, Polishook & White, 2004*; *Rodriguez et al., 2009*; *Cordier et al., 2012*; *Scholtysik et al., 2013*; *Peršoh, 2015*).

Several grass species, such as *Lolium perenne*, are of high agronomic importance and are part of a regular food supply for livestock (*Malinowski & Belesky, 2006*). Such grass species and their interacting symbionts can be influenced by management of grasslands. For example, fertilization and grazing can influence the availability of nutrients for host plants and vegetation structure respectively, and have been shown to change individual fungal abundances, species richness and the microbial community structure of fungal communities in soil (*Donnison et al., 2000*; *Parrent, Morris & Vilgalys, 2006*; *Valyi, Rillig & Hempel, 2015*; *Soliveres et al., 2016*). Thus, land-use intensity may also determine foliar fungal assemblages of meadow or pasture grasses.

The interactions between species within fungal communities are little understood and may include direct and/or indirect competition for plant resources (*Saunders, Glenn & Kohn, 2010*; *Suryanarayanan, 2013*). The systemic *Epichloë* endophytes produce chemical compounds which inhibit growth of pathogenic fungi and generate shifts in below ground subsystems by suppressing the root colonization of mycorrhizal fungi (*Siegel & Latch, 1991*; *Yue et al., 2000*; *Clarke et al., 2006*; *Mack & Rudgers, 2008*; *Kumar & Kaushik, 2012*; *Omacini et al., 2012*; *Vandegrift et al., 2015*). Thus, *Epichloë* endophytes may change the species composition of non-systemic fungal endophytes in grass leaves as well. In this

field study, we ask whether or not the presence of systemic grass endophytes of the genus *Epichloë* changes the species composition of foliar fungal assemblages in a host grass along a land-use intensity gradient in two seasons (spring and summer) and in three geographic regions.

## METHODS AND MATERIALS

### Study sites

The study was conducted on 150 grassland sites within the framework of the Biodiversity Exploratories (http://www.biodiversity-exploratories.de), which includes three distinct regions across Germany. The three study regions; Schwäbische Alb (south-west Germany, ALB), Hainich-Dün (central Germany, HAI) and Schorfheide-Chorin (north-east Germany, SCH), represent different climatic conditions, soil types, landscapes and land-use types, as well as different management intensities (*Fischer et al., 2010*). All selected study sites are real-world grasslands and are not experimental plots (*Fischer et al., 2010*). Rather, they are grasslands used by the owners or farmers to meet their needs without artificial changes by researchers (*Fischer et al., 2010*). Some owners' management strategies have included sowing grasslands with commercial seed mixtures within the last ten years. Such real-world study systems are necessary to show how ecosystems work, but bear the risk of lower replicability compared to controlled laboratory experiments. The grasslands are classified along a land-use intensity gradient (LUI), which integrates the most common practices such as mowing, grazing, and fertilization, into one index, comprising values from zero (extensive) to four (intensive; *Bluethgen et al., 2012*). Intensively managed grasslands are fertilized, grazed by livestock several times during the year and/or mown repeatedly. Extensively managed grasslands, such as semi-natural grasslands, including protected calcareous grasslands and wetlands, are not fertilized and are mown only once and/or grazed for only a short time (*Bluethgen et al., 2012*). For this study we used the LUI calculated for the management in 2014, one year before our sampling in 2015.

Field work permits were issued by the responsible state environmental offices of Baden-Württemberg, Thüringen, and Brandenburg (according to § 72 BbgNatSchG).

### Plant sampling

The perennial ryegrass *Lolium perenne* was selected as the study species, as it is an important forage grass which is commonly associated with the vertically transmitted endophyte *Epichloë festucae* var. *lolii* (formerly *Neotyphodium lolii*; *Klapp & Opitz von Boberfeld, 2013*; *Leuchtmann et al., 2014*). Samples of *L. perenne* were collected in all three study regions in spring and summer surveys in 2015. In total, 80 sites within the 150 grasslands contained *L. perenne* populations and were sampled. In each survey, we sampled up to 20 *L. perenne* plants randomly at different locations at each study site, with a minimum distance of 1 m between sampled plants to reduce the probability of sampling the same plant twice. The number of sampled plants per study site differed depending on the population size of *L. perenne* and on recent mowing and/or grazing events. Overall, 2,147 plants were sampled. Approximately 3 cm of one grass tiller from each plant was collected, and included basal stem, leaf sheaths, and basal leaf blades (*König et al., in press*). The mycelia of *Epichloë*

endophytes accumulate mainly in basal leaf sheaths of the grasses (*Spiering et al., 2005*). The collected basal stem and leaf sheaths of the tillers were therefore used to detect *Epichloë* infections using immunoblot assays (Fig. 1A). As leaf blades contain a high diversity of fungal endophytes (*Sánchez Márquez, Bills & Zabalgogeazcoa, 2007*; *Sánchez Márquez, Bills & Zabalgogeazcoa, 2008*; *Sánchez Márquez et al., 2010*), foliar fungal assemblages of one *Epichloë*- infected and one *Epichloë*- free grass individual per study site and season were assessed in the basal leaf blades (Fig. 1B). The sampled plant material was stored separately for each individual in 2.5 ml Eppendorf reaction tubes. During the field survey, all samples were immediately cooled with dry ice. To prevent degradation of the fungal DNA, plant samples were stored afterwards at −20 °C (*Peay, Baraloto & Fine, 2013*; *Millberg, Boberg & Stenlid, 2015*).

### *Epichloë* endophyte detection

To detect *Epichloë* endophytes in the basal stem and leaf sheaths, a commercially available kit for immunoblot assays was used, following the manufacturer's protocol (http://www.agrinostics.com). In total, 270 (12.6%) of 2,147 sampled *L. perenne* plants were infected with an *Epichloë* endophyte. To compare fungal assemblages of *Epichloë*-infected and *Epichloë*-free samples, 52 sites with *Lolium perenne* which contained both infected and un-infected individuals were chosen from the 80 grassland sites. Depending on recent mowing or grazing events, 21 of the 52 grasslands were sampled exclusively in spring and 15 grasslands exclusively in summer, while 16 grasslands could be sampled in both seasons. As 43% of grasslands contained less than three *Epichloë*-infected plant individuals, one immuno-positive (*Epichloë*-infected, E+) and one immuno-negative (*Epichloë*-free, E−) plant sample for each grassland site and season was randomly selected, resulting in a total of 68 E+ and 68 E− samples.

### Foliar fungal assemblages

For analyses of the foliar fungal assemblages by Next Generation Sequencing (NGS), 68 E+ and 68 E− leaf blades were used. In order to detect the complete foliar (i.e., epi- and endophytic) fungal assemblage on the grass leaves the leaf blades were not surface sterilized.

### DNA extraction

The ChargeSwitch® gDNA Plant Kit (Invitrogen™, Karlsruhe, Germany) was used to extract DNA as recommended by the manufacturer, but with volumes scaled down to 10%. Cell disruption was achieved using a FastPrep®-24 Instrument (MP Biomedicals, Eschwege, Germany) as detailed by *Guerreiro et al. (2018)*.

### Library preparation and sequencing

The fungal barcoding region, i.e., the ITS rRNA gene region, was amplified as detailed by *Guerreiro et al. (2018)*. Briefly, library preparation comprised two sequential amplification steps. In the first PCR, the fungus specific primers ITS1F and ITS4 were used and modified at the 5′-ends to include sample-specific TAG sequences. In the second PCR, the sequencing primers, indices, and the P5 and P7 adapters for the Illumina sequencing were appended. Libraries were processed by the sequencing service of the Faculty of Biology at LMU

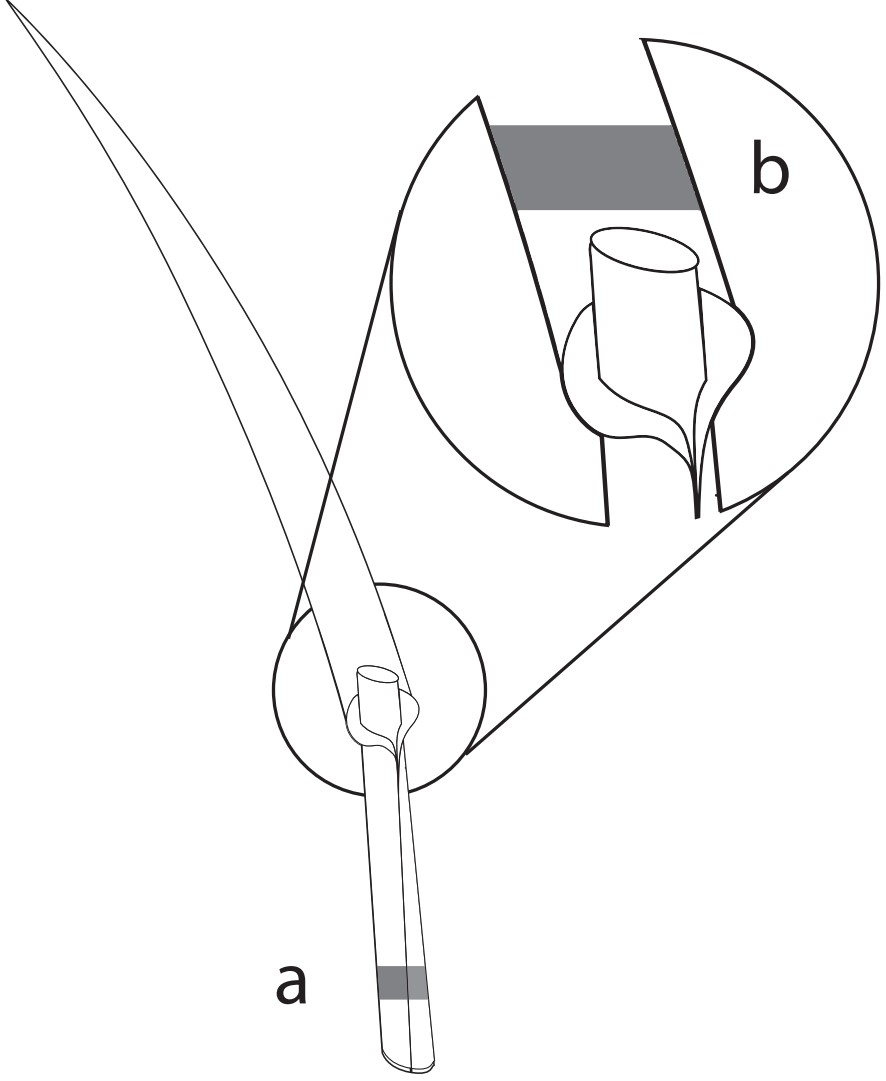

**Figure 1** **Material from the same grass tiller of each sampled *L. perenne* plant was used for immunoblot assays and NGS.** (A) Parts of the basal stem and leaf sheaths were used for detection of *Epichloë* endophyte by immunoblot assays, while (B) parts of the basal leaf blades were used for analyses of the foliar fungal assemblages by NGS.

Munich, and sequenced using an Illumina MiSeq® sequencer (Illumina, Inc., San Diego, CA, USA) with 2 × 250 bp paired end sequencing (MiSeq Reagent Kit v3 Chemistry, Illumina, Inc., San Diego, CA, USA).

## Processing of sequencing data

The obtained sequence reads were processed as detailed by *Guerreiro et al. (2018)*. Briefly, the sequences were demultiplexed using QIIME version 1.9.0 (*Caporaso et al., 2010*). Using the FastX toolkit (http://hannonlab.cshl.edu/fastx_toolkit), reads were trimmed at the 5′-end to comprise only the final 11 bp of the SSU rRNA gene region. These

pre-processed sequence data were deposited in the European Nucleotide Archive database (http://www.ebi.ac.uk/ena/data/view/PRJEB23523). CD-HIT-OTU for Illumina reads. Version 0.0.1 (http://weizhongli-lab.org/cd-hit-otu, *Li et al., 2012*) was selected for clustering reads into Operational Taxonomic Units (OTUs) at a similarity threshold of 97%, according to a previous comparison of the performance of clustering algorithms (*Röhl et al., 2017*). A matrix (OTU table) listing the read count per OTU and sample was generated, which was used for statistical analyses and was deposited in BExIS database (ID 22188; Table S1). The taxonomic affiliation of each OTU was assigned using the UNITE database version 7.1 (*Kõljalg et al., 2005*) as reference.

Samples with less than 15,000 reads were discarded and the OTU read counts were standardized per sample by the total number of reads. Sequence processing revealed 247 fungal OTUs, represented by 4,907,006 quality-filtered ITS1 sequence reads. As eight samples had to be discarded after the ITS sequencing, we ended up with a sample size of 128 (63 E+ and 65 E−) from (a) 19 sites in the region ALB, together 46 samples (spring 24, summer 22); (b) 14 sites in the region HAI, together 30 samples (spring 22, summer 8); and (c) 19 sites in the region SCH, together 52 samples (spring 23, summer 29).

### Statistical analyses

For statistical analyses, the software R version 3.1.1 (*R Development Core Team, 2014*) was used. The effects of the following explanatory variables were tested using linear mixed effect models (LME, nlme package, *Pinheiro et al., 2016*): (i) presence of *Epichloë* endophytes, (ii) season, (iii) region, (iv) land-use intensity on the response variables "species richness" (number of OTUs), "species evenness" and "Shannon diversity" of the OTUs. Study site ID was included as random intercept. Unequal sample sizes for region and season lead to an unbalanced sampling design, but E+ and E− samples were almost equal (one E+, one E−) for each studied grassland. To compare species richness, species evenness, and Shannon diversity between season and region, Tukey's HSD comparison of groups in mixed effects models was used (GLHT, multcomp package, *Hothorn, Bretz & Westfall, 2008*). Differences between the composition of foliar fungal assemblages in dependence of region, season, land-use intensity and *Epichloë* endophyte infection were tested with a PERMANOVA (9,999 permutations) (ADONIS, vegan package, *Oksanen et al., 2017*). It fits a linear model to a distance matrix and tests hypotheses by permutations, thus not assuming normality of the data (*Anderson, 2001*). To characterize compositional differences between foliar fungal assemblages, non-metric multidimensional scaling (NMDS, vegan package, *Oksanen et al., 2017*) based on Bray–Curtis dissimilarities was used. Mean ± SE are used throughout the manuscript unless otherwise specified.

## RESULTS

We identified 247 fungal OTUs associated with 128 *L. perenne* leaves (63 E+, 65 E−) and identified 33 genera. In total, 59% of the OTUs were assigned to Ascomycota, 33% to Basidiomycota and less than 1% to Chytridiomycota (Table 1). Approximately 8% of the OTUs could be not identified at the phylum level (Table 1), which is below the proportion of unassignable fungi in NGS surveys of different habitats such as tree leaves (40%,

**Table 1 Fungal taxa identified in 128 *L. perenne* leaves from 52 grasslands in three German study regions by Next Generation Sequencing of the ITS rRNA gene region.** Shown are the number of OTUs of each order, as well as their percent relative abundance and their proportions of the sequencing reads.

| Phylum | Class | Order | OTUs | % of OTUs | % of reads |
|---|---|---|---|---|---|
| Ascomycota | | | 145 | 58.7 | 42.8 |
| | Dothideomycetes | | 60 | 24.3 | 26.1 |
| | | Capnodiales | 13 | 5.3 | 14.8 |
| | | Dothideales | 1 | 0.4 | 0.1 |
| | | Incertae sedis | 5 | 2.0 | 0.3 |
| | | Pleosporales | 36 | 14.6 | 10.3 |
| | | Unidentified | 5 | 2.0 | 0.6 |
| | Eurotiomycetes | | 9 | 3.6 | 0.2 |
| | | Chaetothyriales | 6 | 2.4 | 0.1 |
| | | Eurotiales | 3 | 1.2 | 0.1 |
| | Lecanoromycetes | | 2 | 0.8 | 0.0 |
| | | Lecanorales | 2 | 0.8 | 0.0 |
| | Leotiomycetes | | 24 | 9.7 | 7.6 |
| | | Erysiphales | 3 | 1.2 | 0.1 |
| | | Helotiales | 21 | 8.5 | 7.6 |
| | Pezizomycetes | | 1 | 0.4 | 0.1 |
| | | Pezizales | 1 | 0.4 | 0.1 |
| | Pezizomycotina | | 4 | 1.6 | 1.1 |
| | | Incertae sedis | 4 | 1.6 | 1.1 |
| | Saccharomycetes | | 2 | 0.8 | 0.0 |
| | | Saccharomycetales | 2 | 0.8 | 0.0 |
| | Sordariomycetes | | 27 | 10.9 | 5.2 |
| | | Hypocreales | 13 | 5.3 | 2.2 |
| | | Sordariales | 4 | 1.6 | 0.2 |
| | | Incertae sedis | 1 | 0.4 | 0.2 |
| | | Xylariales | 9 | 3.6 | 2.6 |
| | Taphrinomycetes | | 2 | 0.8 | 0.1 |
| | | Taphrinales | 2 | 0.8 | 0.1 |
| | Unidentified | | 14 | 5.7 | 2.4 |
| Basidiomycota | | | 82 | 33.2 | 53.9 |
| | Agaricomycetes | | 7 | 2.8 | 0.2 |
| | | Agaricales | 5 | 2.0 | 0.1 |
| | | Polyporales | 1 | 0.4 | 0.0 |
| | | Trechisporales | 1 | 0.4 | 0.0 |
| | Agaricostilbomycetes | | 5 | 2.0 | 0.2 |
| | | Agaricostilbales | 5 | 2.0 | 0.2 |
| | Cystobasidiomycetes | | 3 | 1.2 | 0.3 |
| | | Incertae sedis | 2 | 0.8 | 0.1 |
| | | Unidentified | 1 | 0.4 | 0.2 |

**Table 1** (*continued*)

| Phylum | Class | Order | OTUs | % of OTUs | % of reads |
|--------|-------|-------|------|-----------|------------|
| | Exobasidiomycetes | | 3 | 1.2 | 0.1 |
| | | Entylomatales | 1 | 0.4 | 0.0 |
| | | Unidentified | 2 | 0.8 | 0.1 |
| | Microbotryomycetes | | 19 | 7.7 | 6.4 |
| | | Leucosporidiales | 9 | 3.6 | 1.9 |
| | | Microbotryales | 2 | 0.8 | 0.2 |
| | | Incertae sedis | 1 | 0.4 | 0.0 |
| | | Sporidiobolales | 7 | 2.8 | 4.2 |
| | Pucciniomycetes | | 1 | 0.4 | 0.0 |
| | | Pucciniales | 1 | 0.4 | 0.0 |
| | Tremellomycetes | | 38 | 15.4 | 43.5 |
| | | Cystofilobasidiales | 6 | 2.4 | 2.9 |
| | | Filobasidiales | 6 | 2.4 | 4.4 |
| | | Tremellales | 24 | 9.7 | 34.9 |
| | | Unidentified | 2 | 0.8 | 1.2 |
| | Ustilaginomycotina | | 3 | 1.2 | 0.1 |
| | | Malasseziales | 3 | 1.2 | 0.1 |
| | Wallemiomycetes | | 1 | 0.4 | 0.1 |
| | | Wallemiales | 1 | 0.4 | 0.1 |
| | Unidentified | | 2 | 0.8 | 3.1 |
| Chytridiomycota | | | 1 | 0.4 | 0.1 |
| | Chytridiomycetes | | 1 | 0.4 | 0.1 |
| | | Rhizophlyctidales | 1 | 0.4 | 0.1 |
| Unidentified | | | 19 | 7.7 | 3.3 |

*Yang et al., 2016*), submerged litter (36%, *Röhl et al., 2017*), or dead wood (16%, *Peršoh & Borken, 2017*). The orders contributing the most species to the foliar fungal assemblages of *L. perenne* were the Pleosporales (15% of OTUs), Heliotiales (8% of OTUs) and Hypocreales (5% of OTUs; Table 1). The dominant genera contributing to the foliar fungal assemblages of *L. perenne* were *Cryptococcus* (25% of the sequencing reads), a genus of Basidiomycota, and *Mycosphaerella* (11% of the sequencing reads; Table 2), a genus of Ascomycota.

The sample sizes of analysed *L. perenne* plants differed slightly between the studied regions (ALB = 46, HAI = 30, SCH = 52) and between seasons (spring = 69, summer = 59). Nonetheless, separate species (OTU) accumulation curves for the studied regions (ALB = 216 OTUs, HAI = 205 OTUs, SCH = 220 OTUs) and seasons (spring = 233 OTUs, summer = 215 OTUs; Figs. 2A and 2B) were close to saturation.

The foliar fungal assemblages of *L. perenne* were significantly different among the regions (Table 3), with lowest species richness and highest species evenness in the southernmost and coolest region, ALB (number of OTUs = 51 ± 3, evenness = 0.23 ± 0.004), compared to HAI (number of OTUs = 63 ± 3, evenness = 0.20 ± 0.003) and SCH (number of OTUs = 62 ± 3, evenness = 0.21 ± 0.004), but with no differences in Shannon diversity (ALB = 2.55 ± 0.10, HAI = 2.52 ± 0.10, SCH = 2.63 ± 0.10; Fig. 3). The assemblages in ALB and

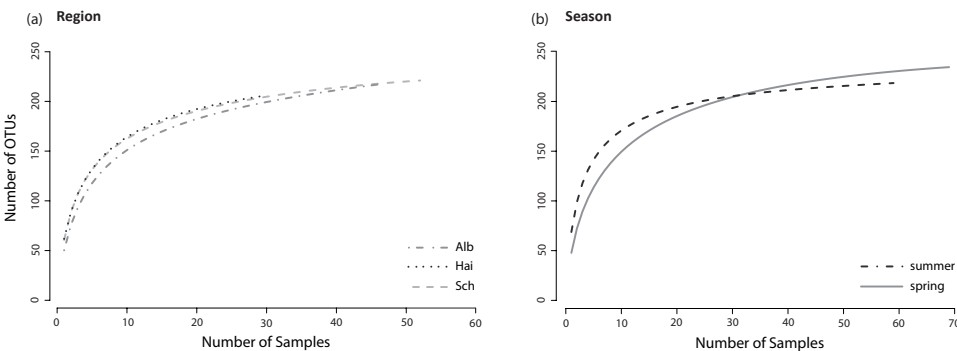

**Figure 2** **Species accumulation curves.** Species accumulation curves of fungal OTUs found in lower leaf blades of *L. perenne* indicate species saturations (A) for each study region: ALB, HAI, and SCH; and (B) for both seasons: spring, summer. All fungal OTUs were included, resulting in asymptotic curves.

**Table 2** **Dominant fungal genera identified in 128 *L. perenne* leaves from 52 grasslands in three German study regions by Next Generation Sequencing of the ITS rRNA gene region.** Shown is the proportion of sequencing reads for each genus. Table includes only genera which account for a minimum of 1% of the sequencing reads and does not include undefined OTUs.

| Phylum | Class | Order | Family | Genus | % of reads |
|---|---|---|---|---|---|
| Basidiomycota | Tremellomycetes | Tremellales | Incertae sedis | Cryptococcus | 24.9 |
| Ascomycota | Dothideomycetes | Capnodiales | Mycosphaerellaceae | Mycosphaerella | 11.1 |
| Basidiomycota | Tremellomycetes | Tremellales | Incertae sedis | Bullera | 4.9 |
| Basidiomycota | Tremellomycetes | Filobasidiales | Filobasidiaceae | Filobasidium | 4.4 |
| Basidiomycota | Tremellomycetes | Tremellales | Incertae sedis | Dioszegia | 4.2 |
| Ascomycota | Dothideomycetes | Pleosporales | Didymellaceae | Neoascochyta | 4.1 |
| Basidiomycota | Microbotryomycetes | Sporidiobolales | Incertae sedis | Sporobolomyces | 4.1 |
| Ascomycota | Leotiomycetes | Helotiales | Incertae sedis | Helgardia | 3.1 |
| Ascomycota | Leotiomycetes | Helotiales | Helotiaceae | Articulospora | 2.6 |
| Ascomycota | Sordariomycetes | Xylariales | Incertae sedis | Monographella | 2.4 |
| Basidiomycota | Tremellomycetes | Cystofilobasidiales | Cystofilobasidiaceae | Itersonilia | 1.1 |
| Basidiomycota | Microbotryomycetes | Leucosporidiales | Leucosporidiaceae | Leucosporidium | 1.1 |
| Ascomycota | Dothideomycetes | Pleosporales | Incertae sedis | Boeremia | 1.0 |
| Ascomycota | Pezizomycotina cls Incertae sedis | Incertae sedis | Incertae sedis | Volucrispora | 1.0 |
| Basidiomycota | Tremellomycetes | Cystofilobasidiales | Incertae sedis | Mrakiella | 1.0 |

SCH were distributed in a loosely scattered pattern, while fungal assemblages in the HAI region were more similar to one another (Fig. 4A).

Season significantly affected species richness and composition of the assemblages (Table 3). Species richness and Shannon diversity were higher and compositions of the foliar fungal assemblages were more similar in summer (number of OTUs = 69 ± 3, Shannon = 2.66 ± 0.10) compared to spring (number of OTUs = 49 ± 2, Shannon = 2.50 ± 0.10). Species evenness, however, peaked in spring (spring: evenness = 0.23 ± 0.004, summer: evenness = 0.21 ± 0.003; Figs. 3 and 4B).

**Table 3 Statistical results.** Effects of *Epichloë* infection (INF), season (SEA), region (REG) and land-use intensity (LUI) on fungal species richness (number of OTUs), evenness, Shannon diversity and fungal composition. Significant *p*-values are highlighted in bold.

| | df | Species richness [r][a] | | Species evenness [J'][a] | | Shannon diversity [H'][a] | | df | Fungal composition[b] | |
|---|---|---|---|---|---|---|---|---|---|---|
| | | F | p | F | p | F | p | | F | p |
| INF | 1,74 | 1.91 | 0.171 | 1.15 | 0.287 | 0.49 | 0.485 | 1,122 | 0.64 | 0.788 |
| SEA | 1,74 | 47.41 | **<0.001** | 20.14 | **<0.001** | 6.01 | **0.017** | 1,122 | 11.45 | **<0.001** |
| REG | 2,48 | 7.63 | **0.001** | 11.99 | **<0.001** | 0.43 | 0.655 | 2,122 | 9.08 | **<0.001** |
| LUI | 1,48 | 2.03 | 0.161 | 0.37 | 0.545 | 2.00 | 0.164 | 1,122 | 1.51 | 0.129 |

**Notes.**
[a]Data were analysed by a linear mixed-effect model with study site ID as random effect.
[b]Data were analysed with PERMANOVA (9999 permutations).

Neither land-use intensity nor infection with *Epichloë* endophytes had a significant effect on species richness (number of OTUs: E+ = 56 ± 3, E− = 60 ± 3) or composition of fungal assemblages of *L. perenne* leaves (Table 3, Figs. 4C and 4D).

From all 63 immunoblot positive (E+) samples of basal stems and leaf sheaths, only 17% (11 samples) indicated the occurrence of *Epichloë festucae* var. *lolii* in lower leaf blades using the NGS method. In 5% (7 of all 128 analysed *L. perenne* samples), NGS detected *Epichloë uncinata*.

# DISCUSSION

Many of the dominant and ubiquitous ascomycetes detected by NGS in our study, including several taxa such as *Acremonium, Alternaria, Cladosporium, Epicoccum* and *Penicillium,* have previously been recorded in other grass species (*Sánchez Márquez, Bills & Zabalgogeazcoa, 2007*; *Sánchez Márquez et al., 2010*) and in *L. perenne* (*Thomas & Shattock, 1986*) with direct isolation methods. In contrast to these studies, the fungal genus that dominated in our study belonged to the Basidiomycota (*Cryptococcus)*. With direct isolation methods, only cultivable fungi can be detected, while indirect methods such as NGS can also detect fungi which cannot be cultured *in vitro*. Such differences in detection probabilities may have resulted the observed differences between our study and those of others. However, the presence of numerous fungal species seems to be characteristic of the mycobiome of grasses, leading to large compositional similarities in comparisons of fungal assemblages from different grass species (*Neubert et al., 2006*; *Sánchez Márquez, Bills & Zabalgogeazcoa, 2007*; *White & Backhouse, 2007*; *Sánchez Márquez, Bills & Zabalgogeazcoa, 2008*; *Sánchez Márquez et al., 2010*).

## Effects of region, season and land-use intensity

The three study regions; ALB, HAI, and SCH, span a latitudinal gradient from south to north across Germany, including different grassland types with variable vegetation structures (*Fischer et al., 2010*). The total number of OTUs was similar between regions, but the fungal assemblages, including mean number of OTUs, differed strongly between the regions in our study. Similarly, other studies have also found differences between regions for fungal assemblages of other grass species (*Wirsel et al., 2001*; *Wilberforce et al., 2003*; *Neubert et al., 2006*; *Sánchez Márquez, Bills & Zabalgogeazcoa, 2008*). The environmental

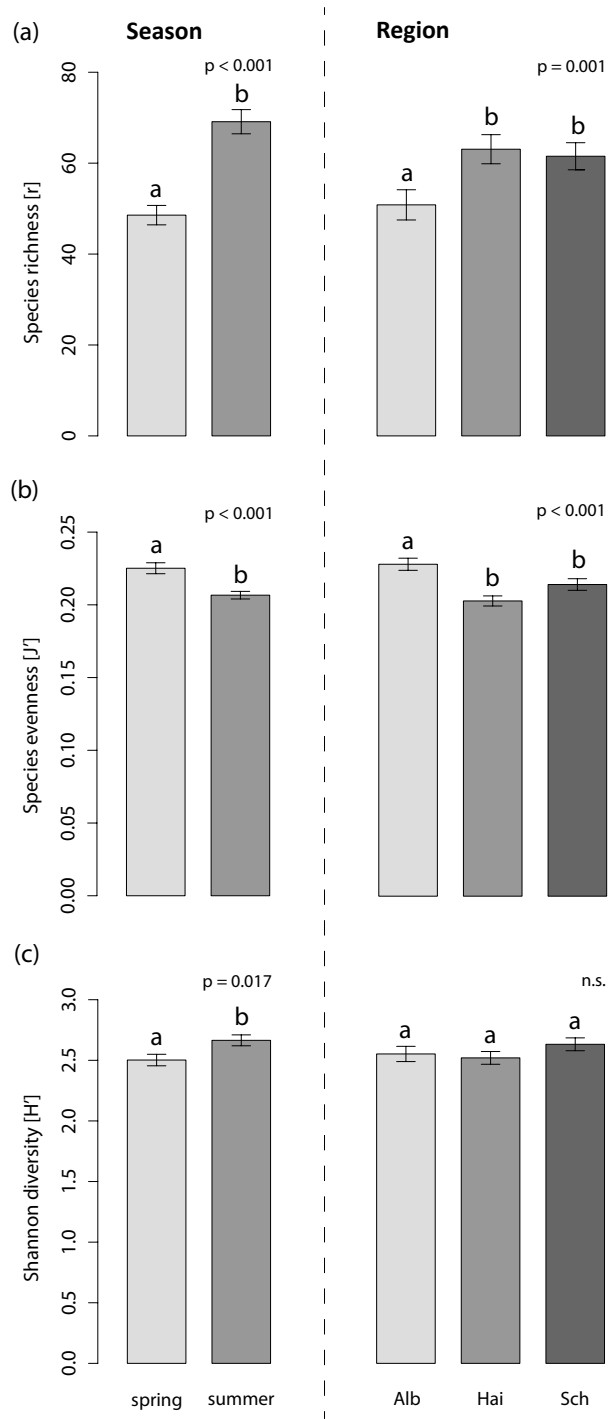

**Figure 3 Effects on the species indices of the foliar fungal assemblages in the grass *L. perenne*.** The (A) species richness (number of OTUs), (B) evenness and (C) diversity of the foliar fungal assemblages in the grass *L. perenne* depending on season and study region. Means ± SE are shown. Different letters above bars indicate significantly different groups at $p < 0.05$, corrected for multiple comparisons.

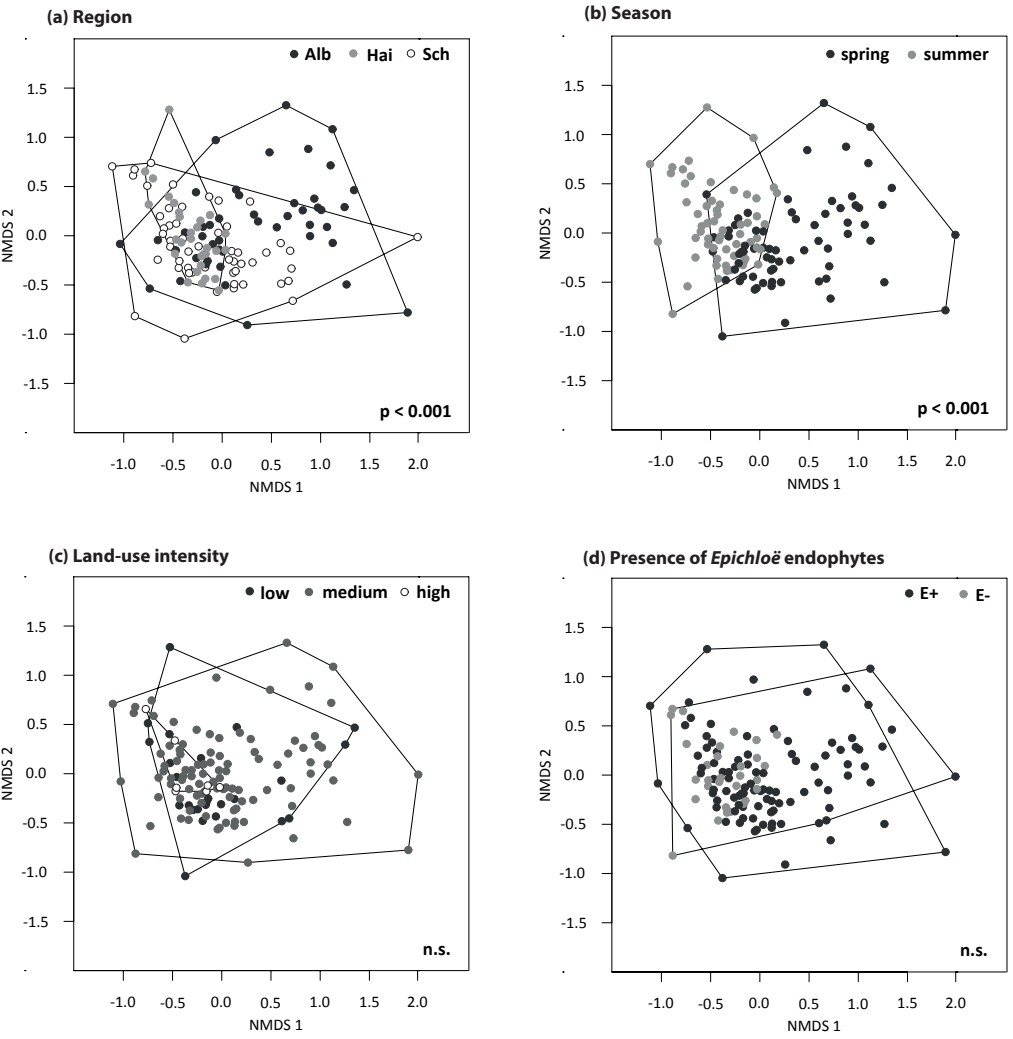

**Figure 4   NMDS ordination (stress = 0.20) of foliar fungal community composition in *L. perenne*.** Relationships with (A) study region, (B) season, (C) land-use intensity and (D) the presence of *Epichloë* infection are highlighted. Dots represent different foliar fungal assemblages of *L. perenne*. Polygons indicate clustering of fungal compositions based on the analysed variables.

context of study sites, including soil, vegetation, surrounding landscape, weather, and climate may contribute to differences in the foliar fungal assemblages among the three regions in our study.

Apart from study region, species richness and species composition changed between spring and summer. Similar seasonal changes have been observed in fungal assemblages of different tree species (*Peršoh, 2015*). This seasonal pattern may be due to the accumulation of aerial and rain-dispersed fungal spores over time (*Sánchez Márquez et al., 2012*). As leaves grow older, susceptibility to infections by horizontally transmitted fungal endophytes increases (*Balazs, Gaborjanyi & Kiraly, 1973*; *Iwasa et al., 1996*); and leaves of temperate grasses tend to die with summer drought (*Sánchez Márquez et al., 2012*). This could explain

the significantly higher species richness and Shannon diversity but lower species evenness in summer as compared to spring.

In contrast to region and season, land-use intensity had a minor impact on foliar fungal assemblages of *L. perenne* in our study. A recent study found that taxonomic richness of different endophytic fungi, including mycorrhizal fungi in roots, decreased with increasing mowing intensity on the same study sites in the three regions (*Simons et al., 2017*). Another study found that both, species richness and diversity, of below ground fungi were negatively affected by increased N mineralization rates, but effects on abundances of different taxa varied (*Parrent, Morris & Vilgalys, 2006*). We assume that different management practices, such as mowing or fertilization, essentially change the likelihood of the occurrence of any single species, but that the overall effect on fungal assemblages remains rather low.

## Effects of *Epichloë* endophytes

While infections with *Epichloë* endophytes have been shown to affect mycorrhizal colonization of grasses (*Mack & Rudgers, 2008*; *Omacini et al., 2012*; *Vandegrift et al., 2015*), they had no significant effect on the foliar fungal assemblages of *L. perenne* in our study. Similar to a recent study on *Festuca rubra* and *Epichloë festucae* (*Zabalgogeazcoa et al., 2013*), neither species richness nor the composition of fungal communities in leaves changed between *Epichloë*-infected and *Epichloë*-free samples. We analysed fungi from the surfaces and internal tissues of *L. perenne* leaves. In addition to endophytic species, fungal epiphytes and spores on the outer surface of the grass leaves were detected in our study. The presence of these epiphytes and spores may have confounded our results somewhat, as we assume that *Epichloë* endophytes have a stronger effect on fungi which had invaded the leaves.

Interestingly, in only 17% of the plants where an infection with *E. festucae* var. *lolii* was detected in the basal stems and leaf sheaths by immunoblot assays (E+), the endophyte species was also detected by NGS in the lower leaf blades. Infection rates of *Epichloë* may differ among plant parts (*Spiering et al., 2005*) and the limited specificity of immunoblot assays may result in false positive results (*Jensen et al., 2011*). *Epichloë* fungal DNA in the host plants increase with plant age (*Fuchs et al., 2017*) and therefore, younger basal stems have a lower detection probability of the fungi compared to older plants. The reason why we achieved such low overlap between the two methods needs further study, as a causal connection could not be established with our study design and sampling methods, using different plant material for the immunoblot assays and NGS.

Furthermore, in some *L. perenne* plants (5%), independent of the *Epichloë* infection detected by immunoblot assays, NGS detected *Epichloë uncinata*, a species not found in *L. perenne* (*Leuchtmann et al., 2014*). Since we sampled *L. perenne* plants at the vegetative stage in heterogeneous and species rich grasslands, we may have sometimes inadvertently sampled the hybrid Festulolium or young meadow fescue tillers (*Festuca pratensis*). Festulolium can be visually difficult to distinguish from *L. perenne* and farmers have seeded this hybrid; it is frequently included in seed mixtures used by managers (J König, pers. comm., 2014). Both Festulolium and *F. pratensis* species can serve as hosts of *E. uncinata*. *Epichloë uncinata* itself is a hybrid of the species *Epichloë bromicola* and the *E.*

*typhina* complex, and *E. typhina* has been recorded in *L. perenne* (*Leuchtmann et al., 2014*). This may explain, at least in part, the detection of *E. uncinata* in our samples.

## CONCLUSION

Our results demonstrated that, in all regions, the leaves of the grass *L. perenne* contain more than 200 taxa of fungal endo- and/or epiphytes. The number of OTUs ranges from 50 to 70 fungal taxa per study site depending on region and season. We therefore conclude that the fungal community composition of the leaves depends on study region and season, while land-use intensity of the grasslands and the occurrence of *Epichloë* endophytes in the grass has a minor influence on the foliar endo- and epiphytes in our study. However, land-use intensity has been shown to drive communities of endophytes (*Parrent, Morris & Vilgalys, 2006*; *Valyi, Rillig & Hempel, 2015*; *Soliveres et al., 2016*; *Simons et al., 2017*) and the occurrence of *Epichloë* endophytes changes the expression of over one third of the host genes (*Dupont et al., 2015*) and can also increase resistance to pathogenic fungi (*Bonos et al., 2005*; *Clarke et al., 2006*; *Vignale et al., 2013*; *Xia et al., 2015*). Further studies are needed to exclude the epiphytes and spores on the leaves (e.g., by surface sterilisation), and to detect effects of land use and *Epichloë* endophytes on the fungal diversity of host plants, with a focus on specific and perhaps competing groups of endophytic fungi in the host.

## ACKNOWLEDGEMENTS

We thank the editor Teri Balser, Maria Julissa Ek Ramos and one anonymous reviewer for helpful comments on our manuscript, Kathleen Regan for a profound native speaker check and Lena Papp and Christopher Sadlowski for assisting in lab work. Andreas Brachmann (Munich) supported library preparation and conducted the Illumina sequencing. We thank the managers of the three Exploratories, Kirsten Reichel-Jung, Katrin Lorenzen, Martin Gorke and all former managers for their work in maintaining the plot and project infrastructure, with special thanks to Uta Schumacher, Steffen Both and Ralf Lauterbach for supporting field work and communication with local farmers. We also thank Christiane Fischer, Jule Mangels, Anja Höck and Cornelia Weist for giving support through the central office, Michael Owonibi for managing the central data base, and Markus Fischer, Eduard Linsenmair, Dominik Hessenmöller, Daniel Prati, Ingo Schöning, François Buscot, Ernst-Detlef Schulze, Wolfgang W. Weisser and the late Elisabeth Kalko for their role in setting up the Biodiversity Exploratories project.

### Funding

The work has been funded by the DFG Priority Program1374 ''Infrastructure-Biodiversity-Exploratories'' (KR 3559/3-1). The publication was funded by the University of Wuerzburg in the funding programme Open Access Publishing. The funders had no role in study design, data collection and analysis, decision to publish, or preparation of the manuscript.

## Grant Disclosures

The following grant information was disclosed by the authors:

DFG Priority Program1374 ''Infrastructure-Biodiversity-Exploratories'': KR 3559/3-1.
University of Wuerzburg.

## Competing Interests

The authors declare there are no competing interests.

## Author Contributions

- Julia König conceived and designed the experiments, performed the experiments, analyzed the data, prepared figures and/or tables, authored or reviewed drafts of the paper, approved the final draft.
- Marco Alexandre Guerreiro performed the experiments, analyzed the data, prepared figures and/or tables, authored or reviewed drafts of the paper, approved the final draft.
- Derek Peršoh performed the experiments, analyzed the data, contributed reagents/-materials/analysis tools, authored or reviewed drafts of the paper, approved the final draft.
- Dominik Begerow analyzed the data, contributed reagents/materials/analysis tools, authored or reviewed drafts of the paper, approved the final draft.
- Jochen Krauss conceived and designed the experiments, contributed reagents/materials/analysis tools, authored or reviewed drafts of the paper, approved the final draft.

## Field Study Permissions

The following information was supplied relating to field study approvals (i.e., approving body and any reference numbers):

Field work permits were issued by the responsible state environmental offices of Baden-Württemberg, Thüringen, and Brandenburg (according to § 72 BbgNatSchG).

## DNA Deposition

The following information was supplied regarding the deposition of DNA sequences:

The pre-processed sequence data (ITS rRNA gene region) were deposited in the European Nucleotide Archive database accession number PRJEB23523.

http://www.ebi.ac.uk/ena/data/view/PRJEB23523.

## Data Availability

The related raw data were deposited on the BExIS database of the Biodiversity Exploratories, ID 22188 and are also available in Table S1.

## Supplemental Information

Supplemental information for this article can be found online at http://dx.doi.org/10.7717/peerj.4660#supplemental-information.

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
