# Peer review of "Knowing your neighbourhood—the effects of Epichloë endophytes on foliar fungal assemblages in perennial ryegrass in dependence of season and land-use intensity"

_PeerJ, doi:10.7717/peerj.4660_

## Round 0.1 · original submission · Major Revisions

The reviewers and myself agree that the paper is a nice contribution, with a novel approach. It merits publication. In order to get to that point, there are a few things to be done.

As both reviewers say (and I concur), the paper has some basic editorial work needed. There are some minor English corrections to be made, and some clarifications needed regarding the methods used. The two reviewers have provided annotated pdf files that should assist in this regard. I would also like to see more of an explanation of why you chose to look at land use change. It seems an added complication that is not fully discussed.

In addition to editorial revision, Reviewer 2 has some very useful suggestions for expanding the results section and deepening the discussion. I agree that these would improve the paper. I am not convinced that the paper needs to be entirely re-written (per reviewer 2), but I do think that some serious consideration to those comments would improve the submission markedly.

Reviewer 1 ·

Basic reporting

Some corrections of the English text are indicated in the revised pdf file.

Experimental design

The main objective of the study is to determine if the presence of Epichloë in plants of Lolium perenne affect the structure of other components of the fungal mycobiota of the plants. To answer this question, an indirect method of fungal detection and identification based on NGS has been used. To my knowledge, this an innovative approach not yet used to test the above hypothesis.
Comments to make more clear the description of some methods have been made in the annotated pdf file.

Validity of the findings

Some remarks to improve the findings of the study:

1. The listing of the fungal taxa in Table 1 is fine, but another list with the dominant genera will be very useful to assimilate the findings, and to compare them with those of other surveys. You might want to add the frequencies of different OTU belonging to the same genus for this purpose.

2. In the first paragraph of the discussion section consider that the method for fungal detection used here was NGS, and not direct isolation like in the cited surveys of grass mycobiota. Similar results are obtained in terms of some dominant taxa, supporting this claim. In contrast with surveys based in direct isolation, a large number of basiomycota (33% of OTU) have been detected using NGS.

3. Provide in the text or in table 2 the species richness values for E+ and E- plants.

4. What is the species richness value used (e.g. in figure 2) ? number of taxa? specify this on the text, figure axis, or figure caption.

Annotated reviews are not available for download in order to protect the identity of reviewers who chose to remain anonymous.

·

Basic reporting

This is a very interesting report on fungal endophyte community assemblages in grasses. it is well written although has some important typos and inconsistencies that should be revised. The structure meets journal standards.

Experimental design

The research question is well defined, relevant and meaningful, however some methods, although well described, do not meet with high technical standards to support the results, discussion and conclusions presented. It is highly recommended the authors rewrite several parts to clarify changes within the methods and materials section that are not consistent. Also, they should include references regarding criteria for sampling design, storage, processing and bioinformatic analysis within the results and discussion section. One suggestion is this article: Conducting a Microbiome Study Goodrich, Julia K. et al.
Cell , Volume 158 , Issue 2 , 250 - 262.

Validity of the findings

My main concern is that the authors do not present solid foundations for their conclusions. The results section is very short and the authors do not discuss putative pitfalls that they might have had during the study, as I can infer based on the inconsistencies presented in the methods and materials section.

Additional comments

This is a very interesting manuscript. This research area needs more information regarding how the microbiome interacts under different conditions. I recommend you rewrite this interesting study focussing on the technical part. I think you presented several inconsistencies based on the high volumen of data you had. Many researchers in this area need to have more systematic ways to design the experiments and to perform meaningful statistical analyses and present them in a very easy way for readers.

---

## Round 0.2 · Minor Revisions

The manuscript is much improved. Reviewer 2 has helpfully suggested some minor editorial changes and a suggested sentence to add that will finalise the paper. Please do check those, and I look forward to seeing your final version!

Reviewer 1 ·

Basic reporting

See below

Experimental design

See below

Validity of the findings

See below

Additional comments

See below

·

Basic reporting

The current version is now clear and unambiguous. I thank the authors for making new figures. Also, the materials and methods section now is very well explained . I detected several typos, please double check before the manuscript is accepted as it is.

Experimental design

Yes, now the current version shows a well defined research question. The materials and methods are well described.

Validity of the findings

I consider their results are valid. However, it has to be stated that their findings were obtained under certain conditions that not necessary are easy to replicate. Please add one sentence commenting this. I do not necessary have to review it.

Additional comments

I consider your manuscript almost ready for publication. I appreciate your consideration to my comments and suggestions. You nicely answer my questions and concerns. Only one last recommendation, please double check for any typo you might still have.

---

## Round 0.3 · accepted · Accept

You have done a great job addressing the reviewers comments, and I am happy to let you know that we can now accept the manuscript. Congratulations!

# ·

Basic reporting

Manuscript is ready for publication.

Experimental design

Manuscript is ready for publication.

Validity of the findings

Manuscript is ready for publication.

Additional comments

Manuscript is ready for publication.